# Analysis of *MTNR1A* Genetic Polymorphisms and Their Association with the Reproductive Performance Parameters in Two Mediterranean Sheep Breeds

**DOI:** 10.3390/ani13030448

**Published:** 2023-01-28

**Authors:** Asma Arjoune, Abrar B. Alsaleh, Safia A. Messaoudi, Hanen Chelbi, Refka Jelassi, Mourad Assidi, Taha Najar, Brahim Haddad, Marc-André Sirard

**Affiliations:** 1Centre de Recherche en Reproduction, Développement et Santé Intergénérationnelle, Département des Sciences Animales, Faculté des Sciences de l’Agriculture et de l’Alimentation, Université Laval, Quebec, QC G1V 0A6, Canada; 2Department of Animal Production, National Agronomic Institute of Tunisia, University of Carthage, Tunis 1082, Tunisia; 3Department of Forensic Sciences, Naïf Arab University for Security Sciences, Riyadh 14812, Saudi Arabia; 4Laboratory of Medical Parasitology, Biotechnologies, and Biomolecules, Pasteur Institute of Tunis, Tunis 1002, Tunisia; 5Center of Excellence in Genomic Medicine Research (CEGMR), Faculty of Applied Medical Sciences, King Abdulaziz University, Jeddah 21589, Saudi Arabia

**Keywords:** *MTNR1A*, SNPs, sheep, reproductive parameters

## Abstract

**Simple Summary:**

The aim of this research was to study the effect of *MTNR1A* gene polymorphisms on the reproductive performance of two Mediterranean sheep breeds. A total of 26 SNPs were found, and three SNPs caused amino acid changes. Two SNPs were totally linked and correlated with reproductive activity resumption. This study consists of a piece of new useful knowledge for the control of sheep reproductive seasonality and farm management that could improve genetic selection, especially in countries where agriculture relies mainly on sheep farming.

**Abstract:**

Sheep farming plays an important economic role, and it contributes to the livelihoods of many rural poor in several regions worldwide and particularly in Tunisia. Therefore, the steady improvement of ewes’ reproductive performance is a pressing need. The *MTNR1A* gene has been identified as an important candidate gene that plays a key role in sheep reproduction and its sexual inactivity. It is involved in the control of photoperiod-induced seasonality mediated by melatonin secretion. The aim of this study was to identify SNPs in the *MTNR1A* gene in two Tunisian breeds, Barbarine (B) and Queue Fine de l’Ouest (QFO). DNA extracted from the blood of 77 adult ewes was sequenced. Selected ewes were exposed to adult fertile rams. A total of 26 SNPs were detected; 15 SNPs in the promoter region and 11 SNPs in the exon II were observed in both (B) and (QFO) breeds. The SNP rs602330706 in exon II is a novel SNP detected for the first time only in the (B) breed. The SNPs rs430181568 and rs40738822721 (SNP18 and SNP20 in our study, respectively) were totally linked in this study and can be considered a single marker. DTL was associated with SNP18 and SNP20 in (B) ewes (*p* < 0.05); however, no significant difference was detected between the three genotypes (G/G, G/A, and A/A) at these two SNPs. Fertility rate and litter size parameters were not affected by SNP18 and SNP20. There was an association between these two polymorphisms and (B) lambs’ birth weights (*p* < 0.05). Furthermore, the ewes with the A/A genotype gave birth to lambs with a higher weight compared to the other two genotypes for this breed (*p* < 0.05). There was not an association between SNP 18 and SNP20 and (QFO) ewes’ reproductive parameters. These results might be considered in future sheep selection programs for reproductive genetic improvement.

## 1. Introduction

The most expressed seasonal rhythm for small ruminants living in the Mediterranean areas is reproductive activity. It is a form of adaptation that several sheep breeds exert in the face of changes in environmental factors [1]. Ewes’ reproduction seasonality is manifested by the existence of the seasonal anestrus period (the period in days from ram placement with ewes to lambing); it is controlled by the photoperiod [2,3,4]. Photoperiod annual variation is one of the main environmental factors that synchronizes reproductive activity and melatonin secretion [5]. Therefore, the short-days season is characterized by shorter photoperiods and promotes nocturnal synthesis of melatonin that stimulates their sexual activity [6]. Furthermore, the melatonin nocturnal production time in fall and winter is longer than in spring and winter [7]. It exerts a fundamental function in the modulation of sheep reproduction rhythm [8] through specific receptors located in the hypothalamic, pituitary, and gonadal axis [9]. Two high-affinity melatonin receptors have been identified in mammals: melatonin receptor 1A (*MTNR1A*, MT1) and melatonin receptor 1B (*MTNR1B*, MT2) [10,11]. *MTNR1A* is located on chromosome 26 in sheep. So far, several scientists have explored polymorphisms on the *MTNR1A* gene. In addition, these mutations were associated with the reproductive seasonality traits in different sheep breeds [12,13,14,15,16,17,18]. However, with *MTNR1B*, although it is less studied, it was reported that polymorphism in this receptor gene could affect ewes’ reproductive seasonality and litter size [19]. Two single nucleotide polymorphisms (SNP) at positions 606 and 612 have been associated with different reproductive parameters [20,21,22]; however, in the Ile de France breed, there is no association between these SNPs and reproductive traits [23]. Discovering a polymorphic site in the *MTNR1A* gene has provided new useful knowledge for the control of sheep reproductive seasonality and farm management that could improve genetic selection, especially in countries where agriculture relies mainly on sheep farming. It will be beneficial to study the *MTNR1A* gene polymorphism in two main sheep breeds in Tunisia, Barbarine (B) and Queue Fine de l’ouest (QFO). Therefore, this research aims at studying *MTNR1A* polymorphisms in two Tunisian sheep breeds and investigating their association with the ewes’ reproductive performance parameters.

## 2. Materials and Methods

### 2.1. Ethical Approval

The care and the use of animals were under the control of the veterinarians of the Livestock and Pastures office (LPO) of Tunisia in accordance with the guidelines of the Animal Ethics Committee of the National School of Veterinary Medicine (Ethical approval no: CEEA-ENMV 57/2022). Blood samples were collected by a veterinarian during routine flock health control procedures. 

### 2.2. Animals and Management

Ewes (*n* = 77) of meat-type breeding were used in this study. The ewes were of two Tunisian autochthonous breeds: Barbarine (*n* = 42) and Queue Fine de l’ouest (*n* = 35). These two breeds are the most typical purebred breeds used in sheep breeding. The animals were produced and located in the northeast of Tunisia (Zaghouan Governorate) at the LPO farms; the farming was based on a semi-extensive system located at a latitude of 36.3667 and a longitude: of 9.9 36°22′0″ North and 9°54′0″ East with a Mediterranean climate and a hot summer. All the ewes selected for the present study were multiparous aged from 3 to 5 years (3.825 ± 0.785) with body condition score (BCS) from 3.5 to 4 and had been maintained in a natural photoperiod condition from birth. The ewes were characterized by seasonal breeding; they show a period of anoestrus from March to September. They can lamb once a year from January to March. In addition, the ewes were clinically examined and their BCS on a 1 to 5 scale was assessed by a veterinarian.

### 2.3. Reproductive Data Collection

Rams of the two breeds were introduced in female flocks on the 1st of May of each year, so lambing can occur in October where there are more grazing areas, ensuring adequate milk production and lamb development. Rams were moved away from the ewe flocks after 90 days. For each breed, days to lambing (DTL), the number of lambs born for each ewe, and lambs’ birth weights were recorded for 2 years/campaigns: 2017/2018 and 2018/2019. From the farm data records, the following reproductive variables for both breeds were collected: DTL, number of lambed ewes, litter size, and lambs’ birth weights.

### 2.4. Sampling and Sequencing

Ten (10) mL of blood from the jugular vein of each ewe was collected using vacutainer collection EDTA tubes. Total DNA was extracted using the DNeasy blood and tissue kit (Qiagen, Mississauga, ON, Canada) according to the manufacturer’s protocol. All the primers used for the amplifications of the promoter and exon II regions were those reported by Luridiana et al. [12]. The polymerase chain reaction (PCR) and the electrophoresis conditions were performed following Luridiana et al. [12]. All PCR products were sequenced in forward direction by a commercial service (Macrogen, Seoul, Korea). Information about primers and fragment amplifications are available in Appendix A.

### 2.5. Sequencing Data Analysis

DNA sequencing results were visualized using Series Data Collection Software 3130 (Thermo Fisher Scientific, Waltham, MA, USA) and analyzed using Sequencing Analysis software v5.3.1 (Thermo Fisher Scientific, USA). Genetic mutations were detected using Seqscape® software v2.6 (Thermo Fisher Scientific, USA) [24]. The sequences alignment was conducted via MEGA X: Molecular Evolutionary Genetics Analysis with the reference sequence of the sheep genome AY524665.1 and NM_001009725 from GenBank.

### 2.6. Statistical Analysis 

Statistical analysis of the results was performed using the software SPSS Statistics version 28 (IBM, New York, NY, USA). Frequencies of genetic changes were compared using the Chi-squares (*X*_2_) test and in case of non-validity the Fisher’s exact tests. Accordingly, comparisons of variables between 2 groups were carried out using ANOVA test for normally distributed variables. As for the continuous variables, they were expressed in these mean ± standard deviation (SD). The value of *p* ≤ 0.05 was considered statistically significant.

## 3. Results

The sequences of *MTNR1A* promoter and exon II regions from the total sample were compared with the reference sequence AY524665.1 and NM_001009725 from GenBank. All genotype positions were inserted according to the sheep genome assembly version Oar_rambouillet_v1.0:26:17354220:17378573 (Oar_Rambouillet_v1.0, accession RefSeq GCF_002742125.1). As shown in Table 1, an overall total of 26 SNPs were detected; 15 SNPs in the promoter region, and 11 SNPs in the exon II were observed in both (B) and (QFO) breed. The SNPs number, breeds identifiers, location, locus, and amino acid change in the MTNR1A gene are summarized in Table 1. The allele and genotype frequency and the Odds Ratio (OR) are included in Table 2. All SNPs were in Hardy–Weinberg equilibrium except SNP6 (rs415456480) in both breeds (*p* = 0.004 and *p* ≤ 0.0001 for (B) and (QFO), respectively) and SNP7 (rs406184829) within the promoter region in (QFO) ewes (*p* = 0.00027) due to their high heterozygosity rate. Within exon II, the most studied region, five SNPs were missense, and the others were synonymous. Three SNPs caused amino acid changes, (rs407388227 G > A and rs404378206 G > A causing Val/Ile and rs416266900 A > C causing Ala/Asp substitution) in the amino acid sequence. The others twenty-three SNPs detected in our study were silent.

The most frequently evaluated SNPs in many ewes’ breeds in previous study (rs406779174, rs430181568, rs407388227) for the association with seasonal reproduction were detected in our study. The SNP rs430181568 and rs407388227 (SNP18 and SNP20 in our study, respectively) were linked (D’ = 0.99 and r^2^ = 0.94), so that they can be considered as a single marker. Figure 1 shows the linkage disequilibrium plot among all the SNPs in *MTNR1A* gene detected in our study.

In the current study we found that the G allele was the most frequent in rs430181568 (SNP18) and rs407388227 (SNP20) loci in the two studied breeds (0.83 and 0.81 for (B) and (QFO), respectively), and consequently, the genotype G/G (70%) was the most frequent. Instead, for rs406779174, C (0.63) and T (0.67) alleles were the most frequent in (B) and (QFO) breeds, respectively (Table 2). Consequently, C/C (44%) and T/T were the most frequent genotypes in (B) and (QFO) breeds, respectively. 

Of the possible 2^26^ haplotypes, 15 haplotypes were found to be common and were included in the subsequent analysis. Haplotype names and frequencies are shown in Only eight haplotypes had a frequency greater than 5% and only five of them were significant (*p* < 0.05) (Appendix A).

Each SNP was analyzed to determine its relationship with DTL, fertility rate, litter size, and lambs’ birth weights in both ewes’ breeds. SNP17 (rs406779174), SNP18 (rs430181568), and SNP20 (rs407388227) were associated with DTL in the second year (2018/2019) (*p* < 0.05) but not in the first year (2017/208) of the study (Appendix A). Furthermore, the ewes with G/G and G/A genotypes at SNP18 and SNP20 had a shorter DTL (*p* < 0.05) (Appendix A). Ewes with TT and CC genotypes at SNP17 had a shorter DTL (*p* < 0.05) (Appendix A).

Additionally, SNP18 (rs430181568) and SNP20 (rs407388227) were associated with lambs’ birth weights in 2019 (*p* < 0.05) in all ewes from both breeds; ewes with G/G and A/A genotypes at SNP18 and G/A and A/A genotypes at SNP20 gave birth to lambs with a better birth weight (*p* < 0.05) (Appendix A). SNP18 and SNP20 were not associated with fertility rate and litter size in all ewes from both breeds in both years (Appendix A). However, SNP17 (rs406779174) was associated with litter size in the second year of the study (2018/2019) (*p* < 0.05) (Appendix A); it was also associated with lambs’ birth weights in both 2018 and 2019 (*p* < 0.05) (Appendix A).

In the (B) ewes breed, the results from the statistical analysis indicated that SNP18 (rs430181568) and SNP20 (rs407388227) were associated with DTL in the first year of the study (2017/2018) (*p* < 0.05) but not in the second year (Appendix A). Ewes with G/G and G/A at SNP18 (rs430181568) and SNP20 (rs407388227) positions had a shorter DTL (*p* < 0.05) (Appendix A). There was also an association between these two polymorphisms and lambs’ birth weights in 2019 (*p* < 0.05). Furthermore, the ewes with the G/A and A/A genotypes gave birth to lambs with a better weight compared to the ewes with G/G genotypes for this breed (*p* < 0.05) (Appendix A). However, SNP17 was not associated with any reproductive traits in the (B) ewes (Appendix A). 

In the ewes of the (QFO) breed, the values of DTL, fertility rate, and litter size parameters in both years were not affected by the SNP18 and SNP 20 (Appendix A). In addition, there was no association between SNP18, SNP20, and lambs’ birth weights (Appendix A). SNP17 was only associated with litter size in the second year of the study (2018/2019) (Appendix A).

## 4. Discussion

The Barbarine and Queue Fine de l’ouest are the dominant breeds in Tunisia and are raised specially for meat production. They are characterized by a metabolic and digestive adaptation to the contrasting environmental conditions. However, a slow genetic improvement has been shown in sheep flocks of these two breeds. Melatonin, which is normally generated in the brain but can also be secreted by granulosa cells, is a known modulator of follicle development, oocyte maturation, and embryo development [25,26]. In sheep, ewes treated by melatonin displayed better follicle development in culture than those from untreated ewes. Furthermore, melatonin modulates gene expression related to steroidogenesis, differentiation, and granulosa cell luteinization [27]. This hormone protects the integrity of oocytes and granulosa cells via its specific receptors *MTRN1A* and *MTRN1B2* by scavenging reactive oxygen species (ROS) and regulating apoptosis-related genes to prevent apoptosis [28,29]. Our recent study showed that silencing of the *MTNR1A* receptor may be involved in the melatonin antiproliferative effect [30]. Another study reported that *MTNR1A* receptor silencing stimulates follicular atresia by increasing the expression of apoptosis-related genes [31] and reducing progesterone production.

The sequencing analysis of the melatonin receptor 1 gene (*MTNR1A*) in (B) and QFO ewes’ breeds revealed 26 SNPs located in the promoter region and exon II. All the SNPs had been already reported in different breeds [12,13,16,20], except SNP rs602330706 in exon II (SNP 26 in our study), which is, to the best of our knowledge, a novel SNP detected for the first time in our study only in the (B) breed. When compared to the Sarda breed and Awassi ewes, only three polymorphic sites causing amino acid changes were identified among our (B) and QFO ewes [12]. The exon II of the *MTNR1A* gene is the most studied because, within its nucleotide sequence, two specific mutations were found at positions 606 and 612 (SNP17 and SNP18 in our study, respectively) that have been associated with seasonality traits. However, these SNPs are synonymous mutations and therefore not causative in the Rasa Aragonesa breed [16], so it was difficult to explain how they influence the reproductive seasonality [13]. The analysis of the (B) and QFO exon II sequence showed that the SNP18 (old 612/MnlI; rs430181568) was associated with the SNP20 (rs407388227), which was coherent with that found in other Mediterranean and European breeds [12,32,33]. This finding suggests that one or both SNPs may be implicated in the reproduction seasonality [13,32]. This amino acid change (Val > Ile) occurred in the fifth transmembrane domain (TM5), in position 220 of the protein chain [34]. TM5 is an essential site for *MTNR1A* function; amino acid change in this domain could be the origin of receptor signaling modification. Furthermore, other amino acid changes in the *MTNR1A* protein sequence at 195, 208, and 211 were considered as responsible for changing the binding capacity of the melatonin receptor. In addition, differences in cAMP inhibition were detected between Val 220 and Ile 220 at the *MTRN1A* receptor; this finding led us to suggest a potential modification in the molecular signal transduction of the melatonin signal in several sheep breeds with different genotypes at SNP20 [35]. Variations in melatonin signal perception may participate in the significant differences in the reproductive traits in our study. 

In our study, SNP17, the most significative polymorphism, was not associated with DTL in both breeds, which is contradictory with findings in the Istrian Pramenka and Rasa Aragonesa breeds [13,15]. However, SNP18 (rs430181568) was associated with DTL for the first year studied only in Barbarine ewes. These results are in disagreement with those of Martinez-Royo et al. [36], where SNP17 was associated with DTL but not SNP18 (rs430181568) in the Rasa Aragonesa sheep breed. Our results also confirmed that SNP18 and SNP20 (rs407388227) are fully linked, and these data are consistent with the findings of several scientists [12,13,20,32]. In the present study, some reproductive traits in the Barbarine breed were affected by SNP18 (rs430181568) and SNP20 (rs407388227). Interestingly, there were no effects on the reproductive functions for the three genotypes in the (QFO) ewes. In the (QFO) breed, no significative effect has been identified for the reproductive traits such as DTL, fertility rate, and litter size at SNP18 (rs430181568) and SNP20 (rs407388227) in the studied period. However, SNP17 was associated with litter size in QFO ewes in the second year of the study. Furthermore, lambs’ birth weights were affected by SNP18 (rs430181568) and SNP20 (rs407388227) in the (B) breed only; ewes with the G/A and A/A genotypes gave birth to lambs with better weights (*p* < 0.05). 

In fact, Barbarine ewes carrying genotype A/A delayed the onset of the reproductive activity compared to the G/G and G/A genotypes. In both breeds, the different DTL recorded in the two years in the three genotypes may be associated with a different sensitivity to the photoperiod. Considering the mating trend in both the years, Barbarine ewes carrying the G/G and G/A phenotypes mated in the first 30 days after ram introduction; however, ewes with the A/A genotype have the most mating within 40 days from the ram introduction. In the (QFO) ewes, the highest number of the ewes carrying the G/G and G/A genotypes mated in the first 40 days after ram introduction. Therefore, the *MTRN1A* receptor seems to influence reproductive activity in sheep, so the different polymorphisms in SNP18 and SNP20 may modulate the melatonin signal transduction, and this could change the ewes’ reproductive response. 

The different *MTNR1A* genotypes seem to affect the melatonin signal transmission and seasonality, so we can suggest that G/G in (B) ewes was less sensitive to the photoperiod. Their reproductive activity resumption was advanced compared to A/A ewes in both the years. In fact, sheep carrying the G/G and G/A genotypes responded earlier to the ram effect as if they were in a less deep anestrous compared to the A/A genotype, so A/A ewes were surely in a deeper anestrous state. However, different results were reported in QFO ewes for both years. For the first year, the A/A genotypes were advanced in reproductive recovery compared to the G/G and G/A genotypes, but for the second year, the G/A ewes were less sensitive to the photoperiod compared to the G/A and A/A ewes. This means that the highest number of G/G and G/A ewes suggesting that their anestrous state was not as deep or as shallow than in A/A ewes. Instead, ewes with the A/A genotype showed the mating peak later than the other genotypes so they need a longer stimulus to onset the reproductive activity. In fact, the depth of anestrous is the most important factor that affects the response of ewes to the ram introduction [37]; DTL length may depend also on the follicle development. In the (B) and (QFO) breeds, litter size and fertility rate were not associated with SNP18 (rs430181568) and SNP20 (rs407388227) in the *MTNR1A* gene sequence. SNP18 and SNP20 were associated with the body weight of lambs at birth for Barbarine ewes in the last year of the study but not with the (QFO) breed. In fact, Barbarine ewes with the A/A genotype recorded the highest body weight of lambs at birth (*p* < 0.05). This finding was also compatible with the results of Mahdi et al., who reported that mutations located in the *MTNR1A* gene sequence could affect lambs’ weights at birth for the Awassi breed [38]. The effect of SNP18 and SNP20 in the (QFO) breed is considered marginal, which is in agreement with what is described by Cosso et al. and Hernandez et al. in the Awassi and Ile de France breeds, respectively [18,23]. 

In the present study, Barbarine ewes with G/G and G/A showed readiness to resume the reproductive activity earlier than the QFO ewes at the ram introduction. Our different results, from the first and the second year of the study, let us suggest that sheep breed, ewes’ age and environmental factors may be the most important triggers that influence the ewes’ response to the photoperiod; this fact could explain the lack of effect of these two mutations (rs430181568 and rs407388227) on QFO ewes’ reproductive performance parameters.

## 5. Conclusions

Our data confirm the presence of polymorphisms in the *MTNR1A* gene sequence in the Barbarine and Queue Fine de l’Ouest breeds. Twenty-six polymorphic sites were detected, three of which caused amino acid changes. The total linkage of SNP18 and SNP20, which are considered as a single marker, revealed that these polymorphisms can influence the reproductive activity resumption in Barbarine ewes with the G/G and G/A genotypes but not in QFO ewes. These results might be considered in sheep selection programs for reproductive genetic improvement. To explain the association more precisely between polymorphisms in the *MTNR1A* gene sequence and the reproductive traits, other studies with a higher number of ewes from both breeds need to be conducted. In addition, it would be useful to study the effect of the several genotypes at SNP18 and SNP20 on ovarian cells to demystify the intricate role of the *MTNR1A* gene at the ovary.

## Figures and Tables

**Figure 1 animals-13-00448-f001:**
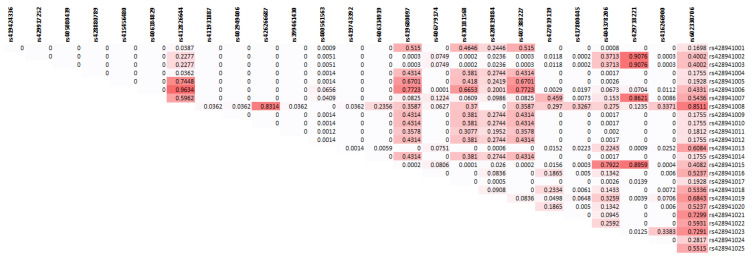
The linkage disequilibrium plot (*p* value) among all the SNPs in *MTNR1A* gene sequence found in our study.

**Table 1 animals-13-00448-t001:** SNPs number, dbSNPs identifiers, location (using the genome version Oar_rambouillet_v1.0 as a reference), *p*-values of the Hardy–Weinberg equilibrium, locus, and amino acid change in the *MTNR1A* gene. B: Barbarine, QFO: queue fine de l’ouest.

SNP Number	dbSNPs	Position	HWE (*p*-Value)			*MTNR1A* Region	A.A′ Change
			B	QFO	All Subject		
SNP1	rs428941001	Chr 26: 17379252	0.52	0.23	0.27	Promoter	–
SNP2	rs419424336	Chr 26: 17379237	1.00	0.31	0.15	Promoter	–
SNP3	rs429917252	Chr26: 17379178	1.00	0.31	0.15	Promoter	–
SNP4	rs405080439	Chr 26: 17379097	0.52	0.76	0.51	Promoter	–
SNP5	rs428880789	Chr 26: 17379083	1.00	0.76	1.00	Promoter	–
SNP6	rs415456480	Chr 26: 17379037	0.004	<0.0001	<0.0001	Promoter	–
SNP7	rs406184829	Chr 26: 17378992	0.19	0.0003	<0.0001	Promoter	–
SNP8	rs412826644	Chr 26: 17378833	1.00	1.00	1.00	Promoter	–
SNP9	rs411931887	Chr 26: 17378874	1.00	0.35	0.67	Promoter	–
SNP10	rs402949406	Chr 26: 17378871	1.00	0.54	1.00	Promoter	–
SNP11	rs426266687	Chr 26: 17378842	1.00	0.54	0.83	Promoter	–
SNP12	rs399461430	Chr 26: 17378769	1.00	0.54	1.00	Promoter	–
SNP13	rs400561563	Chr 26: 17378728	1.00	0.57	0.35	Promoter	–
SNP14	rs419743392	Chr 26: 17378706	1.00	0.36	0.82	Promoter	–
SNP15	rs406334919	Chr 26: 17378624	0.55	0.51	0.15	Promoter	–
SNP16	rs419680097	Chr 26:17355611	0.59	0.13	0.13	Exon2	–
SNP17	rs406779174	Chr 26:17355458	0.20	0.31	0.018	Exon2	–
SNP18	rs430181568	Chr 26:17355452	0.59	0.59	0.27	Exon2	–
SNP19	rs420819884	Chr 26:17355389	1.00	0.17	0.086	Exon2	–
SNP20	rs407388227	Chr 26:17355358	0.95	0.13	0.13	Exon2	Val/lle
SNP21	rs427019119	Chr 26:17355281	0.70	0.57	1.00	Exon2	–
SNP22	rs417800445	Chr 26:17355263	1.00	0.57	0.51	Exon2	–
SNP23	rs404378206	Chr 26:17355190	1.00	1.00	1.00	Exon2	Val/lle
SNP24	rs429718221	Chr 26:17355173	1.00	0.74	0.65	Exon2	–
SNP25	rs416266900	Chr 26:17355171	1.00	0.58	0.72	Exon2	Ala/Asp
SNP26	rs602330706	Chr 26:17355149	1.00	1.00	1.00	Exon2	–

**Table 2 animals-13-00448-t002:** Allele and genotype frequency of the *MTNR1A* gene SNPs in the two sheep breeds. All genotype positions were inserted according to the sheep genome assembly version Oar_rambouillet_v1.0:26:17354220:17378573 (Oar_Rambouillet_v1.0, accession RefSeq GCF_002742125.1).

SNP Number	dbSNPs	Genotype	Genotype Frequency (%)		OR (95% CI)	*p*-Value	Allele	Allele Frequency	
			B	Q				B	Q
SNP1	rs428941001	GG	33 (13)	14 (6)	1.00	0.075	G	0.60	0.44
		GT	54 (21)	60 (25)	2.58 (0.83–7.97)		T	0.4	0.56
		TT	13 (5)	26 (11)	4.77 (1.14–19.98)				
SNP2	rs419424336	CC	72 (28)	45 (19)	1.00	0.018	C	0.85	0.64
		CT	26 (10)	38 (16)	2.36 (0.88–6.29)		T	0.15	0.36
		TT	3 (1)	17 (7)	10.32 (1.17–90.78)				
SNP3	rs429917252	AA	3 (1)	17 (7)	10.32 (1.17–90.78)	0.018	G	0.85 (66)	0.64 (54)
		GA	26 (10)	38 (16)	2.36 (0.88–6.29)		A	0.15	0.36
		GG	72 (28)	45 (19)	1.00				
SNP4	rs405080439	CC	13 (33)	8 (19)	1.00	0.170	C	0.6	0.46
		CT	54 (21)	55 (23)	1.78 (0.62–5.14)		T	0.4	0.54
		TT	13 (5)	26 (11)	3.57 (0.90–14.15)				
SNP5	rs428880789	CC	41 16)	19 (8)	1.00	0.063	C	0.64	0.46
		CT	46 (18)	55 (23)	2.56 (0.90–7.30)		T	0.36	0.54
		TT	13 (5)	26 (11)	4.40 (1.13–17.07)				
SNP6	rs415456480	CC	77 (30)	67 (28)	1.00	0.150	C	0.83	0.7
		CT	13 (5)	7 (3)	0.64 (0.14–2.94)		T	0.17	0.3
		TT	10 (4)	26 (11)	2.95 (0.84–10.34)				
SNP7	rs406184829	AA	87 (34)	69 (29)	1.00	0.056	A	0.92	0.76
		AG	10 (4)	14 (6)	1.76 (0.45–6.84)		G	0.08	0.24
		GG	3 (1)	17 (7)	8.21 (0.95–70.67)				
SNP8	rs412826644	CC	100 (42)	88 (38)	1.00	0.008	C	1	0.94
		CT	0.00	12 (5)	N/A		T	0.00	0.06
		–							
SNP9	rs411931887	AA	14 (6)	28 (12)	5.33 (1.37–20.71)	0.026	G	0.62	0.43
		GA	48 (20)	58 (25)	3.33 (1.10–10.09)		A	0.38	0.57
		GG	38 (16)	14 (6)	1.00				
SNP10	rs402949406	GG	14 (6)	28 (12)	4.86 (1.30–18.13)	0.032	T	0.63	0.44
		TG	45 (19)	56 (24)	3.07 (1.06–8.91)		G	0.37	0.56
		TT	40 (17)	16 (7)	1.00				
SNP11	rs426266687	AA	14 (6)	28 (12)	4.57 (1.22–17.16)	0.05	G	0.62	0.44
		GA	48 (20)	56 (24)	2.74 (0.94–7.98)		A	0.38	0.56
		GG	38 (16)	16 (7)	1.00				
SNP12	rs399461430	AA	40 (17)	16 (7)	1.00	0.032	A	0.63	38
		AG	45 (19)	56 (24)	3.07 (1.06–8.91)		G	0.37	0.56
		GG	14 (6)	28 (12)	4.86 (1.30–18.13)				
SNP13	rs400561563	GA	22 (9)	33 (14)	1.72 (0.65–4.59)	0.27	A	0.89	0.83
		GG	78 (31)	67 (28)	1.00		G	0.11	0.17
		–							
SNP14	rs419743392	CC	38 (15)	14 (6)	1.00	0.038	C	0.61	0.43
		CT	48 (19)	57 (24)	3.16 (1.03–9.70)		T	0.39	0.57
		TT	15 (6)	29 (12)	5.00 (1.28–19.53)				
SNP15	rs406334919	AA	75 (30)	43 (18)	1.00	0.0051	A	0.86	0.63
		AG	22 (9)	40 (17)	3.15 (1.16–8.53)		G	0.14	0.37
		GG	2 (1)	17 (7)	11.67 (1.33–102.72)				
SNP16	rs419680097	GG	70 (30)	70 (30)	1.00	0.88	G	0.83	0.81
		GT	26 (11)	23 (10)	0.91 (0.34–2.46)		T	0.17	0.19
		TT	5 (2)	7 (3)	1.50 (0.23–9.63)				
SNP17	rs406779174	TT	19 (8)	49 (21)	1.00	0.0014	T	0.37	0.67
		TC	37 (16)	37 (16)	0.38 (0.13–1.11)		C	0.63	0.33
		CC	44 (19)	14 (16)	0.12 (0.04–0.41)				
SNP18	rs430181568	GG	70 (30)	70 (30)	1.00	n/a	G	0.83	0.83
		GA	26 (11)	26 (11)	1.00 (0.38–2.66)		A	0.7	0.7
		AA	5 (2)	5 (2)	1.00 (0.13–7.57)				
SNP19	rs420819884	GG	95 (41)	77 (33)	1.00	0.024	G	0.98	0.86
		GA	5 (2)	19 (8)	4.97 (0.99–25.01)		A	0.02	0.14
		AA	0.00	5 (2)	n/a				
SNP20	rs407388227	GG	70 (30)	70 (30)	1.00	0.88	G	0.83	0.81
		GA	26 (11)	23 (10)	0.91 (0.34–2.46)		A	0.17	0.19
		AA	4 (2)	7 (3)	1.50 (0.23–9.63)				
SNP21	rs427019119	AA	9 (4)	0.00	n/a	0.041	A	0.28	0.16
		AG	37 (16)	33 (14)	0.69 (0.28–1.71)		G	0.72	0.84
		GG	53 (23)	67 (29)	1.00				
SNP22	rs417800445	AA	4 (2)	0.00	n/a	0.14	A	0.24	0.15
		AG	40 (17)	30 (13)	0.61 (0.25–1.50)		G	0.76	0.85
		GG	56 (24)	70 (30)	1.00				
SNP23	rs404378206	GG	98 (42)	79 (34)	1.00	0.0042	G	0.99	0.9
		GA	2 (1)	21 (9)	11.12 (1.34–92.14)		A	0.01	0.1
		–							
SNP24	rs429718221	TT	16 (7)	9 (4)	0.44 (0.11–1.83)	0.51	T	0.42	0.34
		TC	51 (22)	49 (21)	0.74 (0.30–1.86)		C	0.58	0.66
		CC	33 (14)	42 (18)	1.00				
SNP25	rs416266900	AA	5 (2)	0.00	n/a	0.14	A	0.23	0.14
		AC	37 (16)	28 (12)	0.60 (0.24–1.51)		C	0.77	0.86
		CC	58 (25)	72 (31)	1.00				
SNP26	rs602330706	CC	95 (41)	100 (43)	1.00	0.093	C	0.98	1
		CT	5 (2)	0.00	n/a		T	0.02	0.00
		–							

## Data Availability

The data presented in this study are available on request from the corresponding author. The data are not publicly available to preserve privacy of the data.

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
