# Peer review of "Analysis of MTNR1A Genetic Polymorphisms and Their Association with the Reproductive Performance Parameters in Two Mediterranean Sheep Breeds"

_animals, 2023, doi:10.3390/ani13030448_

Round 1

Reviewer 1 Report

The manuscript entitled “Analysis of MTNR1A genetic polymorphisms and their association with the reproductive performance parameters in two Mediterranean sheep breeds” summarized MTNR1A gene polymorphisms on the reproductive performance of two Mediterranean sheep breeds. Generally, the manuscript is well written. This paper has several weaknesses and needs improvement before publication.

This manuscript has minor language problems. There are too many for me to modify them all. Authors are strongly encouraged to seek a native English speaker who may assist you modifying the document.

Comments:

1.     Why did the authors choose this technology instead of other GWAS? Please provide the reason.

2.     What cell type was utilized for RNA extraction

3.     How much RNA was used in PCR? Was it one-step or 2-step PCR?

4.     The abstract is not particularly informative and would benefit from more background.

5.     Summarize the abstract, focus on the main findings and mention the small conclusion in at the end of abstract

6.     In the Introduction focus on the objectives and insert a few new reference and relevant findings

7.     In material and method sections, references are missing.

8.     Most of the references mentioned are old and I suggest adding recent references, and the manuscript should be edited accordingly.

9.     I suggest the cite following paper in introduction part For more information you can read below reference

RNA-Seq reveals the potential molecular mechanisms of bovine KLF6 gene in the regulation of adipogenesis. International Journal of Biological Macromolecules, 195, 198-206.

Expression of the bovine KLF6 gene polymorphisms and their association with carcass and body measures in Qinchuan cattle (Bos Taurus). Genomics, 112(1), 423-431.

Bioinformatics analysis and genetic polymorphisms in genomic region of the bovine SH2B2 gene and their associations with molecular breeding for body size traits in qinchuan beef cattle. Bioscience Reports, 40(3).

Detection of polymorphisms in the bovine leptin receptor gene affects fat deposition in two Chinese beef cattle breeds. Gene, 758, 144957.

Association between FASN gene polymorphisms ultrasound carcass traits and intramuscular fat in Qinchuan cattle." Gene 645 (2018): 55-59.

Polymorphism of the PLIN1 gene and its association with body measures and ultrasound carcass traits in Qinchuan beef cattle. Genome63(10), 483-492.

10.  Material and method needs to clarifying and summarizing- some detailed needs

11.  The subtitles in the material and method needs to summarizing Ethical approval and references must be mentioned in M&M

12.  In the result section, the results were written very poorly. It should be written again and try to avoid a brief introduction in the starting of every result.

In conclusion, the research presented is interesting, well planned and carried out. The manuscript can still be improve revise. Nevertheless, I believe that this work deserves publication in after the inclusion of corrections.

Author Response

Dear editors and reviewers,

We would like to thank you and the reviewers for the insightful and constructive comments that have improved the quality of the manuscript.

Herein, please find a new submission of an improved version of the manuscript (with track changes) including some of the suggested comments along with the authors’ answers to the reviewers’ comments.

Our point-by-point responses to the comments of each reviewer are detailed and uploaded to him/her in separate files.

Thank you for your consideration, and we look forward to hearing from you at your earliest convenience.

Sincerely,

Reviewer 2 Report

Manuscript Number:  animals- 2127669

Title: Analysis of MTNR1A genetic polymorphisms and their association with the reproductive performance parameters in two Mediterranean sheep breeds

General comments and judgment

The study of the MTNR1A receptor is certainly an interesting topic that deserves the appropriate recognition in the research field for the improvement of reproductive performance in sheep. However, some points should be improved and clarified, as specified in the section below.

My general judgment is “to be reconsidered after major revision”.

Main concerns

Please remember that the gene nomenclature requires using italics for genes and QTL names, consequently please change MTNR1A with MTNR1A whenever it appears in the text.

M and M section lacks important information, essential for a correct interpretation of the results. Some of the most important concerns are summarized in the following questions: how old were the used ewes? were they primiparous or pluriparous? how is the reproductive cycle of the two breeds treated? do they show a period of anoestrus and if so how long does it last? how many rams were introduced? how was the male/female ratio? were the rams young or “experienced”? had they already produced offspring in previous mating seasons?

Please try to answer all these questions in order to provide the requested information.

The most concern of the study is the Results section. In general, I’m not very sure that tables and text refers to the same number, I suggest an accurate re-examination of the Statistical Analysis applied, pointing more attention to the data set evaluation.

Some aspect may benefit from the attention points listed below.

It is not clear what the authors mean by "correlation". In fact, they declare (Line 164) that SNPs 18 and 20 are "correlated” to the DTL in the second year, and also express a statistical value (p<0.05) but immediately afterwards they say that there is no difference between the 3 found genotypes (165-167). This is not logical, and above all it is not clear how they then obtained the p-value. Reference to the Supplementary Tables 2 and 3 does not help, since non-significance is reported in these. Perhaps the authors meant that there is a difference in days but that this is not significant, but then how do they explain the p<0.05 in Line 165? The same goes for B breed, Lines 170 onwards.

Another concern is on Supplementary Table 4: are the authors sure that a birthweight of 5.9 in 2019 for A/A genotype is less significative than 4.15?

SNP 17 is mentioned in Line 164 along with SNPs 18 and 20, but is not reported in any analysis or Table, why?

Results section need a complete revision before acceptance.

Minor comments

Line 24: please correct the word “heep” in “Sheep”.

Lines 18, 28, 30….and throughout the entire text: pleas use the Italics for gene names.

Line 35: Please carefully check the reference sequence of the SNPs (rs for SNP 20 is wrong) and conform to those present in Tables and in the text.

Line 19 and Line 36: “SNP”, “DTL”, please explain all the acronyms in their first appearance in the text.

Lines 50-51: the definition of the “seasonal anestrus” is not very relevant to the real meaning of anoestrus which is rather a period of sexual inactivity. It would rather appear that you have defined the duration of pregnancy. Please rephrase a definition more responsive to the meaning of aestrus.

Line 155: please delete “of” before “greater”.

Tables

Please change the word “Promotor” with “Promoter” in Table 1 and wherever it appears in the text.

Table 1: Please specify according to  which sequence the localization of the SNPs is established, in particular it is not clear from SNP 16 onwards what the localization refers to and why a unique localization was not followed.

Other minor comments are not included in the present rview report, but I suggest careful check of the entire munuscript, before its re-submission.

Author Response

(The authors gave the same response as above.)

Round 2

Reviewer 2 Report

I would thank the authors for kindly replying to all my comments. The manuscript for me is visibly improved and can be used for publication in its present form.